# Emerging Therapies in Management of Cholangiocarcinoma

**DOI:** 10.3390/cancers16030613

**Published:** 2024-01-31

**Authors:** Jessica Speckart, Veronica Rasmusen, Zohray Talib, Dev A. GnanaDev, Amir A. Rahnemai-Azar

**Affiliations:** 1School of Medicine, California University of Science and Medicine, Colton, CA 92324, USA; jessica.speckart@student.cusm.edu (J.S.); veronica.rasmusen@student.cusm.edu (V.R.); 2Department of Medicine, Arrowhead Regional Medical Center, California University of Science and Medicine, Colton, CA 92324, USA; zohray.talib@cusm.edu; 3Department of Surgery, Arrowhead Regional Medical Center, Colton, CA 92324, USA; 4Division of Surgical Oncology, Department of Surgery, Arrowhead Regional Cancer Center, California University of Science and Medicine, Colton, CA 92324, USA

**Keywords:** cholangiocarcinoma, biliary tract cancer, targeted therapy, precision medicine, genomic aberration, molecular targeted therapies, immunotherapy, genomics, systemic therapy

## Abstract

**Simple Summary:**

This paper delves into the latest developments in the emerging therapies for cholangiocarcinoma, with a focus on precision medicine, immunotherapy, and targeted therapies. We explore the potential of genomic profiling to tailor treatments to individual patients, as well as the promising results of immunotherapeutic agents in clinical trials. Furthermore, we examine the role of specific molecular targets and their inhibitors in slowing disease progression. The findings discussed in this paper offer hope for improved outcomes and prolonged survival in cholangiocarcinoma patients using more effective and personalized treatment strategies.

**Abstract:**

Cholangiocarcinoma is a heterogeneous group of biliary tract cancers that has a poor prognosis and globally increasing incidence and mortality. While surgical resection remains the only curative option for the treatment of cholangiocarcinoma, the majority of cancers are unresectable at the time of diagnosis. Additionally, the prognosis of cholangiocarcinoma remains poor even with the current first-line systemic therapy regimens, highlighting the difficulty of treating locally advanced, metastatic, or unresectable cholangiocarcinoma. Through recent developments, targetable oncogenic driver mutations have been identified in the pathogenesis of cholangiocarcinoma, leading to the utilization of molecular targeted therapeutics. In this review, we comprehensively discuss the latest molecular therapeutics for the treatment of cholangiocarcinoma, including emerging immunotherapies, highlighting promising developments and strategies.

## 1. Introduction

Cholangiocarcinoma (CCA) is a malignancy of the biliary tract with a heterogeneous group of subtypes and a notably poor prognosis. Based on its anatomical origin, it is classified as intrahepatic (iCCA), perihilar (pCCA), or distal (dCCA), with perihilar and distal CCAs often being categorized together as extrahepatic cholangiocarcinoma (eCCA). Over 90% of CCAs, regardless of their subtype, are adenocarcinomas [1]. Historically considered as a rare malignancy, both the incidence and mortality of CCA have risen globally, and CCA now accounts for 3% of gastrointestinal malignancies and 2% of all cancer-related mortalities [2,3]. The high lethality of this disease can be attributed to its clinically silent presentation, aggressive nature, and resistance to chemotherapy.

### Current Management of CCA

Surgical resection is the best treatment option for long-term survival; however, up to 75% of patients are not eligible for resection at the time of presentation due to metastatic or locally aggressive disease [3]. Even after resection, the post-surgical relapse of CCA is common. Therefore, for the majority of CCA patients, systemic chemotherapy is the first-line treatment option. The currently established first-line systemic chemotherapy is cisplatin plus gemcitabine, with years of robust data to support a survival advantage afforded by this therapy [4]. Despite this survival advantage, the prognosis of CCA remains poor, highlighting the need to advance current therapeutic options. 

Recent advances in the identification of oncogenic drivers and the development of targeted molecular therapeutics have created new opportunities for the precision therapy of various cancers, including cholangiocarcinoma (Figure 1). The genetic profiling of CCA has opened a new avenue for targeted treatments, offering the possibility to personalize therapy for patients’ individual tumor biomarkers, which we will discuss in this review. We will outline the current developments and the emerging molecular targets in the treatment of CCA, shedding light on the future of CCA treatment.

## 2. Emerging Therapies

### 2.1. FGFR Inhibitors

The fibroblast growth factor receptor (FGFR) family is a group of receptor tyrosine kinases that participate in a variety of vital functions, such as cellular proliferation, migration, and survival through mitotic and antiapoptotic pathways [6]. Of the four members (FGFR 1–4), FGFR2 aberrations have been identified as the oncogenic drivers of the pathogenesis of cholangiocarcinoma. Most commonly, FGFR2 fusions may be found in 10–15% of CCAs and are currently one of the most promising therapeutic targets for the precision treatment of CCA [7]. 

The FDA has currently approved three FGFR inhibitors for previously treated, unresectable, locally advanced, or metastatic cholangiocarcinoma: pemigatinib, futibatinib, and infigratinib [8,9,10]. Pemigatinib is a selective inhibitor of FGFR 1, 2, and 3 that received FDA approval following the results of the phase II FIGHT-202 clinical trial. In this trial, 38 of 107 patients with FGFR2 fusions or rearrangements achieved an objective response following pemigatinib treatment [11]. A post hoc analysis of the FIGHT-202 study determined that patients who received pemigatinib as their second-line treatment were associated with longer median progression-free survival than those who received systemic second-line treatment prior to pemigatinib targeted therapy [12]. These results support the use of pemigatinib as a second-line treatment for patients with FGFR2 fusions or rearrangements. An ongoing phase II study involving Chinese patients showed similar favorable results, with a 50% objective response rate (ORR) and mean progression-free survival (PFS) of 6.3 months [13]. Following these results, a phase III FIGHT-302 trial is currently underway, investigating the safety and efficacy of pemigatinib as a first-line treatment for patients with unresectable or metastatic cholangiocarcinoma with FGFR2 fusion or rearrangement versus the current first-line systemic therapy of gemcitabine plus cisplatin [14]. Pemigatinib, like other FGFR inhibitors, is considered safe and relatively well tolerated, with hyperphosphatemia as the most common adverse effect. A reduced dose is recommended for patients with severe hepatic or renal impairment [15].

Another inhibitor of FGFR 1, 2, and 3 is infigratinib. Infigratinib received its FDA approval following the results of a phase II clinical trial of 122 patients that demonstrated a 23.1% ORR in treatment-resistant CCA patients with FGFR2 fusions or rearrangements [16]. Similar to pemigatinib, infigratinib has a manageable adverse event profile, as the most common events are hyperphosphatemia, followed by stomatitis, fatigue, and alopecia. However, further clinical trials have been terminated, as the company has decided to discontinue infigratinib development for oncology studies.

Futibatinib, an irreversible inhibitor of FGFR 1–4, has also shown promising results as a second-line treatment for CCA in clinical trials. FOENIX-CCA2, a phase II study, demonstrated a 42% ORR, in 43 of 103 patients, with a median PFS of 9 months, as well as a manageable adverse effect profile [17]. The preclinical studies of futibatinib show evidence that futibatinib is less susceptible to resistance-conferring mutations and more effective against a variety of FGFR2 mutations than pemigatinib and infigratinib [18]. These findings support the further investigation of the efficacy of futibatinib, particularly compared to other FGFR inhibitors, and in cases of the non-fusion or rearrangement FGFR mutations of CCA.

Other FGFR inhibitors currently worth evaluating include derazantinib, erdafitinib, and ponatinib. Derazantinib is an inhibitor of FGFR 1–3 that demonstrated favorable activity in a phase I/II trial with a 20.7% ORR [19]. The clinical investigation of derazantinib is currently ongoing following these encouraging results, and the sponsor has expanded the access to derazantinib for patients with locally advanced, inoperable, or metastatic iCCA with FGFR alterations while phase II studies continue. Following a phase II study that demonstrated a 40% ORR to erdafitinib in patients with locally advanced, unresectable, or metastatic urothelial carcinoma with FGFR alterations, erdafitinib received its FDA approval for the treatment of urothelial carcinoma in 2019 [20]. Erdafitinib’s antitumor activity against FGFR alterations proves that the drug may be a promising option for CCA. A phase IIa study is being conducted in China, investigating the efficacy of erdafitinib against cholangiocarcinoma, as well as non-small cell lung cancer, urothelial cancer, and esophageal cancer (Table 1). Currently indicated for the treatment of chronic myeloid leukemia and Philadelphia chromosome-positive acute lymphoblastic leukemia, ponatinib is a multi-tyrosine kinase inhibitor that targets all members of the FGFR family. A pilot study of ponatinib in patients with FGFR alterations included 12 patients, 1 of whom demonstrated a partial response [21]. Though these results indicate the minimal clinical benefit of ponatinib, the study is limited by its small study size. 

### 2.2. IDH Inhibitors

Isocitrate dehydrogenase (IDH) is a Krebs cycle enzyme responsible for catalyzing the conversion of isocitrate to alpha-ketoglutarate. The mutations of IDH are associated with an abnormal DNA methylation, and, in turn, uncontrolled cell proliferation. Isocitrate dehydrogenase 1 (IDH1) mutations can be found in approximately 13% of iCCA patients [11].

FDA approved ivosidenib in August 2021 for the treatment of locally advanced unresectable or metastatic iCCA, and ivosidenib is a targeted inhibitor of mutated IDH1 [22]. Previous phase I studies have demonstrated that ivosidenib is a safe and well-tolerated drug. Its most common adverse effects are fatigue, nausea, diarrhea, abdominal pain, decreased appetite, and vomiting [23]. The phase III ClarIDHy study successfully demonstrated the significant improvements in the median progression-free survival and the overall survival of patients with IDH1-mutant CCA when given ivosidenib versus a placebo [11,24]. Of the 185 enrolled patients, 124 were assigned to the ivosidenib therapy and 61 received the placebo. The median PFS improved to 2.7 months for the ivosidenib group compared to 1.4 months for the placebo group. However, developing resistance is an inevitable barrier for the long-term IDH1 inhibitor therapy. Secondary IDH1 mutations and IDH1/2 isoform switches have been documented to confer ivosidenib resistance [25]. These mechanisms of resistance provide a rationale for the further development of IDH 1 and 2 inhibitors, such that a sequential strategy of IDH inhibitor therapeutics may be utilized.

Several drugs have been identified as IDH inhibitors and are in various stages of development. Olaparib was evaluated in a phase II study as a monotherapy for patients with IDH1/2 mutation-positive solid tumors [26]. Partial responses were documented in patients with chondrosarcoma and pulmonary epithelioid hemangioendothelioma, but not in any of the four patients with cholangiocarcinoma. Due to the study’s small sample size of CCA patients, further studies are warranted to determine the efficacy of olaparib, especially in combination with other targeted therapies. There are currently five ongoing phase II clinical trials, none of which have posted their results, studying olaparib both as a monotherapy and in combination with other drugs such as the PD-1 inhibitors pembrolizumab and durvalumab and ATR kinase inhibitor ceralasertib (Table 1). As a subject of high research interest, olaparib has the potential to be a significant addition to the treatment of IDH1/2 mutation-positive CCA.

Dasatinib is a multi-kinase inhibitor currently used for the treatment of chronic myeloid leukemia and Philadelphia chromosome-positive acute lymphoblastic leukemia [27] Through novel bioinformatics methods, two separate research groups identified dasatinib as a promising drug candidate to target IDH mutated cholangiocarcinoma [28,29]. A completed phase II trial of dasatinib for patients with both IDH 1 and 2 mutated cholangiocarcinoma demonstrated no clinical benefit in the eight patients that were studied (NCT02428855). No other studies are currently being conducted on dasatinib, despite its bioinformatically identified clinical significance.

The oral drug, LY3410738, is currently being investigated in phase I studies for both solid and hematological cancers with IDH mutations. For cholangiocarcinoma, both IDH1 and 2 mutated cancers will be studied, and LY3410738 will be administered as a monotherapy, in combination with systemic gemcitabine and cisplatin therapy, and combination with the PD-1 inhibitor durvalumab (NCT04521686). Results are not yet available for this study.

### 2.3. VEGF Inhibitors

Tumors create the necessary vasculature for growth through the process of angiogenesis. Angiogenesis is induced by signaling tyrosine kinases called vascular endothelial growth factors (VEGF) that stimulate VEGF receptors (VEGFR). Given the pivotal role of VEGF in angiogenesis, it has garnered interest as a target for cancer treatment [30]. In cholangiocarcinoma, VEGF overexpression is prevalent, with rates of 53.8% in intrahepatic cholangiocarcinoma (iCCA) and 59.2% in extrahepatic cholangiocarcinoma (eCCA) [31].

Apatinib, a tyrosine kinase inhibitor that targets VEGFR-2 receptors, has been investigated in a phase II trial (NCT03251443) involving 26 CCA patients who had previously undergone chemotherapy. Administering apatinib once daily resulted in an overall response rate (ORR) of 11.5% and a median overall survival (OS) of 9 months (95% CI: 4.6–13.4) (Mao et al., 2021) [32]. Another phase II trial (NCT03521219) examined 24 patients with advanced iCCA who had failed gemcitabine-based therapy. The study found that apatinib, given once daily, led to a median progression-free survival (PFS) of 95 days (95% CI: 79.70–154.34 days) and a median OS of 250 days (95% CI: 112.86–387.14 days). The disease control rate for these participants was 62.5% overall (95% CI: 112.86–387.14 days) [33]. Apatinib holds promise as a second-line treatment option for advanced CCA, offering a favorable side effect profile. Additional studies could be useful in determining whether the effects of apatinib could be boosted using transarterial chemoembolization (TACE) or concurrent immunotherapy. 

Ramucirumab is another VEGFR-2 inhibitor, and in a phase II trial (NCT02520141) of 61 advanced, unresectable, pre-treated patients with biliary tract cancer (62% with iCCA and 16% with eCCA), it exhibited median PFS and OS rates of 32% (95% CI: 0.22–0.46) and 58% (95% CI: 0.47–0.72), respectively. Ramuricumab was well tolerated but did not result in an increase in PFS as compared to the established BTC chemotherapy regimens (Lee et al., 2022) [34]. These results are similar to what was found in a larger, multi-site phase II trial (NCT02711553), which studied the addition of ramucirumab to the standard chemotherapy in BTC patients [35].

Surufatinib is a small-molecular inhibitor of VEGFR 1, 2, and 3, as well as FGFR1 and CS1FR. A single-arm, multicenter phase II study (NCT02966821) assessed 39 patients with unresectable or metastatic BTC (including 74.4% with iCCA and 12.8% with eCCA), most having stage IV BTC. A total of 33 patients were included in a 16-week PFS set, resulting in a 16-week PFS rate of 46.33% (95% CI: 24.38–65.73). None of the patients achieved a complete or partial response. The median PFS was 3.7 months (95% CI: 2.4–6.0), and the median PS was 6.9 months (95% CI: 5.1–8.1). These results indicate that surufatinib exhibits a moderate clinical efficacy against CCA and continues to be evaluated in a phase III study (NCT03873532) with a larger sample group. No new adverse effects were documented, and the frequency of significant safety risks such as hemorrhage, impaired liver function, and hypertension was similar to what was found in previous trials of surufatinib [36]. 

The inhibition of VEGF has also been tested in a vaccine format. OCV-C01 is an HLA-A*24:02-restricted three-peptide cancer vaccine targeting VEGFR1, VEGFR2, and KIF20A. A phase II study showed that four out of six advanced BTC patients exhibited vaccine-specific T-cell responses to one or more of the three antigens, with log-rank tests finding that these T-cell responses contributed significantly to the overall survival [37]. Cancer vaccines are still at the early stages of investigation for CCA but should be further investigated as a novel therapy. 

### 2.4. EGFR Inhibitors

EGFR is a tyrosine kinase in the human epidermal growth factor receptor (HER) family. EGFR overexpression is linked to the pathogenesis of cancer, by enhancing cell growth and proliferation, and is estimated to be found in 27.4% of iCCA patients and 19.2% of eCCA patients [31].

The TreeTopp Study was a placebo-controlled phase II trial (NCT03093870) spanning 56 sites that assessed varlitinib treatment plus capecitabine in advanced biliary tract cancer (including patients with iCCA, eCCA, gallbladder cancer, and carcinoma of ampulla). A total of 127 patients were enrolled, 64 of whom received varlitinib plus capecitabine and 63 who received placebo plus capecitabine. The median PFS was 2.83 months for the varlitinib group and 2.79 months for the placebo group (HR, 0.90; 95% CI: 0.60–1.37; *p* = 0.63), and the median OS was 7.8 months for the varlitinib group versus 7.5 months for the placebo group. These results indicate that varlitinib did not significantly improve outcomes compared to the placebo group. However, it is noteworthy that varlitinib exhibited good tolerability, and a further investigation is warranted, particularly in female patients who are more likely to exhibit HER overexpression [38]. The most frequent treatment-related adverse effects (TRAEs) were nausea, an increased blood bilirubin, and diarrhea.

In a phase II trial (NCT01206049) involving patients with advanced biliary tract cancer (including gallbladder adenocarcinoma, iCCA, and eCCA), the combination of gemcitabine-based chemotherapy with either panitumumab (an EGFR inhibitor) or bevacizumab (a VEGF inhibitor) was studied. Out of the 88 patients, 45 received panitumumab with gemcitabine-based chemotherapy, resulting in an impressive overall response rate (ORR) of 45% (95% CI: 29–62), a progression-free survival (PFS) of 6.1 months (95% CI: 5.8–8.1), and a median overall survival (OS) of 9.5 months (95% CI: 8.3–13.3). Researchers concluded that neither panitumumab nor bevacizumab in addition to the combination chemotherapy conferred a significant advantage over the combination chemotherapy alone. It is worth noting that the ORR for the panitumumab group was significantly higher than the bevacizumab group (45% vs. 20%), so panitumumab may still have the potential of being a neoadjuvant treatment [39].

### 2.5. HGF/MET Inhibitors

Hepatocyte growth factor (HGF) receptor, also called mesenchymal–epithelial transition factor (MET or c-MET), is another receptor tyrosine kinase that has been identified as a target for therapy due to its implication in the tumorigenesis and metastasis of certain cancers, including cholangiocarcinoma [40]. MET alterations are relatively rare, with MET amplifications occurring in around 2% of intrahepatic CCA cases [41]. MET inhibitors have demonstrated success, particularly in the treatment of non-small cell lung cancer, and therefore, despite their rare use in CCA, they may be a promising therapy for its subset of cancers.

Despite a 2014 phase I study that showed tivantinib to be safe and tolerable and that it produced a partial response in 20% of patients, further investigations of tivantinib for CCA have not been pursued [42]. Another experimental drug that also showed a tolerable safety profile and demonstrated partial responses in phase I trials is meresitinib [43]. In these dose escalation and confirmation studies, treatment-emergent adverse events of Grade 3 or higher were increased AST, ALT, blood alkaline phosphatase, and hyponatremia. Since then, a phase II study has been conducted, assessing the safety and efficacy of meresitinib in addition to the first-line cisplatin–gemcitabine therapy. While the drug was maintained to be safe and tolerable, no improvement in the mean progression-free survival was achieved in comparison to the standard systemic therapy [35]. 

Capmatinib is a tyrosine kinase inhibitor against c-MET, recently approved for the treatment of metastatic non-small cell lung cancer with MET mutations [44]. Though capmatinib is yet to be studied in clinical trials for CCA, individual case studies have shown partial responses to capmatinib in patients with MET fusions and amplifications [45,46]. These findings justify a further investigation of capmatinib for cholangiocarcinoma.

Tyrosine kinase inhibitors that primarily target other pathways have also been shown to have activity in CCAs with MET mutations. For example, crizotinib and ceritinib (discussed in the ROS1/ALK section of this paper) have both demonstrated efficacy against MET-mutated cancers [47,48]. Cabozantinib, previously discussed as a VEGFR2 inhibitor, was also shown to benefit patients with MET amplifications [49]. MET inhibitors are still in the early stages of development for the treatment of CCA, especially when compared to other molecular targets. However, several drugs and preliminary studies show a potential of treating CCAs with MET aberrations.

### 2.6. ROS1/ALK Inhibitors

ROS1 is an oncogene that encodes a receptor tyrosine kinase involved in the activation of various signaling pathways, including SH2 domain tyrosine phosphatases, ERK1/2 (mitogen-activated protein kinase), IRS-1 (insulin receptor substrate), PI3K (phosphatidylinositol 3-kinase), AKT, STAT3, and VAV3 pathways. In a 2011 study by Gu et al. [50], they found ROS tyrosine kinase activity may be involved in the growth and survival of CCA cancer cells, making ROS1 inhibitors potentially useful in controlling disease. Furthermore, ROS1 mutations have been identified in 31.7% of iCCA patients, making it an attractive therapeutic target [51].

Ceritinib is an ALK inhibitor that targets insulin-like growth factor receptor 1 (IGF-1R) and c-ROS oncogene 1 receptor (ROS1). It has been shown to provide benefit in non-small lung cancer patients with ALK mutations and recently was studied in a phase I trial (NCT02227940) to determine its usefulness in patients with advanced solid tumors. Among the 21 patients enrolled, 7 had CCA and received ceritinib in combination with gemcitabine-based chemotherapy. Among the five evaluable CCA patients, three showed clinical benefits including one patient who exhibited a complete response that lasted 10.3 months. Researchers did not determine a significant benefit from the addition of ceritinib to traditional chemotherapy regimens, but it is important to note this study had a very limited sample size due to the number of evaluable patients. Ceritinib has a manageable toxicity profile, with the most common TRAEs being GI distress, neutropenia, and thrombocytopenia [48].

### 2.7. PI3K/PTEN/AKT/mTOR Signaling Pathway Inhibitors

The PI3K/AKT/mTOR signaling pathway is activated by EGFR and has a role in cancer cell growth and resistance in CCA to chemotherapy. Promisingly, researchers have demonstrated that PI3K/mTOR inhibitors can decrease the proliferation in CCA cell lines with aberrant PI3K/mTOR activity by inducing G1 arrest [52]. The frequency of PI3K/PTEN/mTOR overexpression in iCCA has been established as 4.4% [53]. This establishes PI3K/PTEN/AKT/mTOR as a viable target for CCA treatment.

The EVESOR trial, a phase 1 trial (NCT01932177), focused on the combination treatment of everolimus (an mTOR inhibitor) and sorafenib (a VEGF inhibitor) in 43 cancer patients with various solid tumors, including a subset of 23.2% with CCA. The overall response rate was 6.3% with a disease control rate of 75%. Intriguingly, the CCA patients displayed the longest median progression-free survival (PFS) of 9.9 months, indicating that the combination of sorafenib and everolimus holds the most significant potential efficacy for CCA treatment amongst the studied cancers [54].

In a phase II trial (NCT02631590), copanlisib, a PI3K inhibitor, was studied with gemcitabine and cisplatin in 24 advanced biliary tract cancer patients (patients with unresectable CCA or gallbladder cancer). Of the 19 evaluable patients, they had a median OS of 13.7 months (95% CI: 6.8–18.0 months) and a median PFS of 6.2 months (95% CI: 2.9–10.1 months). The most common treatment-related adverse events were anemia, increased lipase, hypertension, neutropenia, and thrombocytopenia. Although the addition of the PI3K inhibitor did not appear to provide significant clinical benefits or enhance the effects of gemcitabine/cisplatin, it is worth noting that only two patients in this study had documented PI3KCA mutations. Notably, these two patients achieved a stable disease and a partial response, respectively. Thus, more research is warranted on the use of PI3K inhibitors, specifically in patients with PI3K mutation [55].

### 2.8. RAS/RAF/MEK/ERK Signaling Pathway Inhibitors

The mitogen-activated protein kinase (MAPK) signaling pathway is composed of several proteins, most notably RAS and RAF, whose mutations are involved in the pathogenesis of cancer [56]. The most successful attempts to target this pathway focus on the mutations of BRAF. BRAF mutations occur in around 1–5% of CCAs, with BRAF V600E as the most common mutation of this gene [41]. Vemurafenib, a selective inhibitor of BRAF V600E kinase that already has an established efficacy in the treatment of BRAF V600E mutation-positive metastatic melanoma, demonstrated an anecdotal partial response in a 2015 phase II study [57]. Further clinical trials are yet to be conducted for vemurafenib use in CCA; however, in case studies, the drug has been shown to achieve a response in treatment-resistant iCCA with BRAF V600E mutations [58]. Vemurafenib has shown some potential for treating non-melanoma solid tumors, and therefore, may be a promising target for further studies of cholangiocarcinoma [59].

Another inhibitor of V600E-mutated BRAF is dabrafenib. Dabrafenib has been studied in combination with the MEK inhibitor trametinib in the phase II ROAR clinical trial. The drug combination achieved a 47–51% response rate (20–22 out of 43 patients) in patients with BRAF V600E-mutated biliary tract cancer [60]. MEK inhibitors, such as trametinib, also act on the MAPK pathway. In the mouse models of iCCA, a concurrent trametinib administration with a PD-1 inhibitor was shown to optimize survival and treatment efficacy [61]. There is evidence to show that MEK inhibitors are a favorable option for combinatorial treatment, exhibiting strong synergistic effects [62].

### 2.9. Other

Several other pathways are also implicated in the tumorigenesis of cholangiocarcinoma, with varying degrees of dedicated investigation, that may offer a future therapeutic potential and are noteworthy. These include, but are not limited to, Hedgehog, Notch pathways, Wnt/beta-catenin signaling, CK-2 inhibitors, and the JAK/STAT cytokine pathway.

The role of aberrant Hedgehog signaling in the unique cancer biology of cholangiocarcinoma is considered controversial, and therefore, there is no current consensus on the therapeutic potential of targeting this pathway to treat CC [63]. Currently, no clinical trials are studying Hedgehog pathway inhibitors for CCA. 

Notch pathways, on the other hand, have been implicated in the pathogenesis of unique molecular subtypes of cholangiocarcinoma [24,64]. Crenigacestat is a potent Notch inhibitor that has demonstrated in vitro ability to reduce iCCA progression through both VEGF and TGF-beta pathways [65,66]. However, in a phase Ib dose escalation study, crenigacestat was poorly tolerated in patients when given in combination with a first-line systemic therapy [67]. The majority of treatment-emergent adverse events were Grade 3 or higher, and over 50% of patients experienced gastrointestinal disorders. Crenigacestat’s safety and efficacy as a monotherapy have not yet been evaluated, and other Notch inhibitors are not yet in clinical development.

Wnt/beta-catenin signaling is modulated by DKK1, which has been shown to promote tumor progression [13]. DKN-01-a has been developed to target DKK1 and studied in a phase I trial in combination with the systemic gemcitabine/cisplatin therapy [68]. The drug showed a tolerable safety profile, but no measurable improvement in outcomes compared to the existing systemic therapy. An ongoing phase II clinical trial is currently studying DKN-01 in combination with the PD-1 inhibitor nivolumab for the treatment of advanced biliary tract cancer (NCT04057365).

Casein kinase 2 (CK2) is a serine/threonine protein kinase whose increased activity contributes to the malignant and proliferative cell potential in various cancers, including cholangiocarcinoma [69]. Silmitasertib, an oral CK2 inhibitor, has been shown to improve the median progression-free survival when given in combination with the gemcitabine–cisplatin therapy, compared to the standard gemcitabine–cisplatin therapy alone [70]. The drug was also well tolerated and is a potential therapeutic that can be added to the first-line treatment of cholangiocarcinoma.

### 2.10. Immunotherapy: PD-1/PDL-1 Inhibitors

CCA tumors employ various mechanisms to suppress the body’s antitumor T cell response, so many researchers have turned to immunotherapy as a new direction in CCA treatment. These mechanisms include the production of chemokines that suppress T cell extravasation, the generation of reactive nitrogen species, and the increased expression of apoptosis inducers. One of these apoptosis-inducing ligands is programmed death ligand 1 (PD-L1), which is produced by the tumor and results in the decreased proliferation and death of T cells [71]. Notably, the tumor expression of PD-L1 and PD-1 in CCA has a prevalence ranging from 35% to 94% and is associated with a poor prognosis in eCCA [72,73]

In 2022, durvalumab, a PD-1 inhibitor, was the first immunotherapy drug that FDA approved for advanced biliary tract treatment following the TOPAZ-1 placebo-controlled, global multi-center, phase 3 clinical trial (NCT03875235). This trial included 685 participants spanning 105 sites in 17 countries, with a patient population determined to be a representative of general patients with advanced biliary tract cancer. They found a significant increase in survival by adding durvalumab to the gemcitabine-based chemotherapy as compared to placebo plus chemotherapy [74]. As a result, durvalumab is now included in the NCCN regimen for unresectable cholangiocarcinoma.

Toripalimab is an anti-PD-1 monoclonal antibody that has shown benefits in urologic cancer, melanoma, and gastric cancer. In a phase II trial (NCT03951597), toripalimab was combined with lenvatinib (VEGFR, FGFR, PDGFRa, RET, and KIT inhibitor) and GEMOX chemotherapy to assess if PD-1 inhibitors can provide benefit in a pure iCCA population. The study enrolled 30 patients with advanced iCCA and reported an overall response rate (ORR) of 80%, a median overall survival (OS) of 22.5 months (95% CI: 15.6–29.3 months), and a median progression-free survival (PFS) of 10.2 months (95% CI: 9.3–16.8 months). Additionally, patients strongly positive for PD-L1 (46.7% of patients) exhibited higher overall response rates. The combination of toripalimab, lenvatinib, and GEMOX therapy showed greater benefit compared to pairwise combinations and is currently being investigated in another phase II trial (NCT04211168) and will soon be researched in a phase III trial (NCT05342194) due to its promising results for iCCA patients [75]. 

The BilT-01 phase II trial (NCT03101566) studied nivolumab plus gemcitabine/cisplatin therapy as well as nivolumab (anti-PD1) plus ipilimumab (anti-CTLA4) in 64 patients with advanced biliary cancer (62.8% had iCCA) who have not received systemic therapy. While nivolumab seemed to be well tolerated, there was no significant benefit in median PFS, OS, or ORR from adding nivolumab to the standard gemcitabine-based chemotherapy [76]. 

Camrelizumab is a humanized PD-1 IgG4 monoclonal antibody that has shown success in lymphoma, hepatocellular carcinoma, and lung cancer and is beginning to be tested in CCA. In a phase II, multicenter trial (NCT03092895), advanced BTC (84.8% iCCA) patients received camrelizumab plus oxaliplatin-based chemotherapy, resulting in an overall ORR of 16.3%, a median PFS of 5.3 months (95% CI:3.7–5.7), and a median OS of 12.4 months (95% CI: 8.9–16.1). Due to its favorable side effect profile and moderate efficacy, camrelizumab remains a drug of interest in supplementing standard chemotherapy [77]. 

Bintrafusp alfa is a bifunctional fusion protein that exhibits action against both PD-L1 and TGF-beta. In a multicenter phase II trial (NCT03833661), bintrafusp alfa was tested in 159 BTC patients who had previously undergone platinum-based chemotherapy. The median progression-free survival was 1.8 months (95% CI: 1.7–1.8 mo), and the median OS was 7.6 months (95% CI: 5.8–9.7 mo). There was one treatment-related death due to hepatic failure; however, the safety profile, in general, was similar to the previous phase I trials and was manageable. Results show that bintrafusp alfa may have some clinical activity and provide a basis for the continued research on bifunctional proteins for BTC patients [78].

**Table 1 cancers-16-00613-t001:** Clinical trials for CCA and their statuses. Clinical trials for cholangiocarcinoma (many of which include other biliary tract cancers and solid tumors in their study populations) are listed here, organized by clinical research phase and ClinicalTrials.gov identifier. Advanced disease refers to locally aggressive, metastatic, and/or surgically unresectable tumors.

Treatment	Target	Population	Phase	ClinicalTrials.gov Identifier	Status
Ivosidenib	IDH1	advanced solid tumors including CCA	I	NCT02073994 [79]	active
TRK-950 in combination with selected anti-cancer treatment regimens	CAPRIN-1 + multiple drug targets	advanced solid tumors including CCA	I	NCT03872947 [80]	active
Pemigatinib OR Ivosidenib + gemcitabine/cisplatin	FGFR2 + IDH	advanced CCA	I	NCT04088188 [81]	active
LY3410738	IDH1/2	advanced solid tumors including CCA	I	NCT04521686 [82]	active
CAR-macrophages (CT-0508)	CAR macrophages	HER2-overexpressing solid tumors including CCA	I	NCT04660929 [83]	active
MIV 818 + Lenvatinib OR Pembrolizumab	liver-targeting prodrug of TRX-MP	HCC and iCCA	I/II	NCT03781934 [84]	active
Durvalumab + Tremelimumab + gemcitabine/cisplatin	PDL-1 + CTLA4	iCCA	I/II	NCT04989218 [85]	active
Futibatinib	FGFR2	advanced solid tumors including CCA	I/II	NCT05727176 [86]	active
IMM2902	HER2 + CD47	HER2-expressing advanced solid tumors including CCA	I/II	NCT05805956 [87]	active
Entrectinib	TrkA/B/C + ROS1 + ALK	solid tumors with TRKA/B/C, ROS1, or ALK gene rearrangements including CCA	II	NCT02568267 [88]	active
Erdafitinib	FGFR	non-small cell lung cancer, urothelial cancer, esophageal cancer, and CCA	II	NCT02699606 [89]	active
Olaparib	IDH1/2	glioma, CCA, and other solid tumors	II	NCT03212274 [90]	active
Olaparib + Ceralasertib	IDH1/2 + ATR kinase	advanced solid tumors including CCA	II	NCT03878095 [91]	suspended, pending data analysis
Olaparib + Durvalumab	IDH + PD-L1	glioma and CCA	II	NCT03991832 [92]	active
DKN-01 + Nivolumab	DKK1	advanced BTC	II	NCT04057365 [93]	active
Pemigatinib after SBRT	FGFR2	iCCA	II	NCT04088188 [81]	active
Toripalimab + Lenvatinib	PD-1 + multikinase inhibitor	advanced BTC	II	NCT04211168 [94]	active
Infigratinib	FGFR	advanced solid tumors including CCA	II	NCT04233567 [95]	active
Pemigatinib	FGFR2	advanced CCA	II	NCT04256980 [96]	active
AZD6738 + Durvalumab	ATR + PDL-1	BTC	II	NCT04298008 [97]	active
Pembrolizumab + Olaparib	PD-1 + PARP	advanced CCA	II	NCT04306367 [98]	active
Camrelizumab + Apatinib	PD-1 and VEGFR2	advanced iCCA	II	NCT04454905 [99]	active
Zanidatamab	HER2	advanced BTC	II	NCT04466891 [100]	active
Atezolizumab + Derazantinib	FGFR2	advanced iCCA	II	NCT05174650 [101]	active
Durvalumab + GemCis	PDL-1	resectable iCCA	II	NCT05672537 [102]	active
Futibatinib	FGFR2	advanced CCA	II	NCT05727176 [86]	active
SMT-NK + Pembrolizumab OR Pembrolizumab monotherapy	allogeneic natural killer cells + PD-1 inhibitor	advanced BTC	II/III	NCT05429697 [103]	active
Pemigatinib + gemcitabine/cisplatin	FGFR2	advanced CCA	III	NCT03656536 [104]	active
Pembrolizumab + gemcitabine/cisplatin	PD-1	advanced BTC	III	NCT04003636 [105]	active
Futibatinib	FGFR2	advanced iCCA	III	NCT04093362 [106]	active
Toripalimab + Lenvatinib + GEMOX	PD-1 + multikinase inhibitor	unresectable iCCA	III	NCT05342194 [107]	not yet recruiting
Ivosidenib	FGFR2	previously treated CCA	IIIb	NCT05876754 [108]	active

Legend: ATR kinase, Ataxia telangiectasia and Rad3-related kinase; BTC, biliary tract cancer; CAR, chimeric antigen receptor; HER2, human epidermal growth factor receptor-2; PARP, poly (ADP-ribose) polymerase; SBRT, stereotactic body radiation therapy; TRK, tropomyosin receptor kinase; and TRX-MP, troxacitabine monophosphate.

## 3. Conclusions

Surgical resection remains the only potentially curative option for cholangiocarcinoma, but advances in genomic profiling of CCA are allowing for the development of molecularly targeted therapies that may be more effective than gemcitabine-based chemotherapy which has limited success. Clinical trials are beginning to turn up more and more promising results and drugs with proven efficacy over standard treatment are continuing to be FDA-approved every year such as PD-L1 inhibitors (durvalumab), FGFR inhibitors (pemigatinib, futibatinib, and infigratinib), and IDH inhibitors (ivosidenib). Additional studies are still necessary to optimize the use of these targeted therapies, explore combination approaches, and overcome resistance mechanisms. Larger studies are also needed to shed more light on pure populations of cholangiocarcinoma which have unique biology and pathophysiology as compared to other biliary tract cancers which it is often grouped with. The future of CCA treatment appears hopeful, with research offering personalized and more effective therapies for patients with this aggressive malignancy.

## 4. Expert Opinion

Due to CCA’s often-advanced stage at presentation, the lack of discernible diagnostic biomarkers, and the absence of effective systemic therapies, patients with CCA face a dismal prognostic outlook. With a more nuanced comprehension of cholangiocarcinogenesis and its landscape of molecular alterations, the recent advances in genomic profiling and next-generation sequencing techniques have forged a promising path for the management of CCA. However, it is imperative to acknowledge that the translational impact of genomic studies remains constrained by the inherent limitations intrinsic to CCA research. The majority of studies tend to encompass a broad spectrum of advanced biliary tract and pancreatic cancers, “all-comer” studies. Moreover, depending on the anatomical origin of CCA, each subtype manifests a discernable genomic profile, tumor biology, clinical behavior, and prognosis. In addition to these challenges, the high grade of intra- and inter-tumoral heterogeneity exacerbates the already intricate nature of CCA. A heightened complexity arises potentially from a confluence of factors. Potentially due to this resultant complexity arising from a confluence of these factors, despite encouraging results in preclinical studies, the outcomes of clinical trials with the established targeted therapies, such as anti-EGFR medications, have been discouraging. Currently, the forefront of optimism resides in medications designed to target FGFR2 fusion and IDH1/2 mutations as well as immunotherapy, offering a substantial promise for the advancement of future CCA management paradigms.

Designing biomarker-driven cohorts focused on more homogeneous groups of patients is the pivotal determinant in establishing precision medicine in CCA. Additionally, a promising avenue for overcoming the limitations outlined involves innovative treatment strategies that combine targeted therapies with diverse systemic modalities, such as immunotherapy, or aim at various aberrant pathways. Recently, several investigators have underscored the significance of aberrations in both the epigenetic regulation and the miRNA modulation of genes in the pathogenesis of CCA. A better comprehension of the association between epigenetic aberrations and cholangiocarcinogenesis is another promising avenue for the development of novel therapeutics.

## Figures and Tables

**Figure 1 cancers-16-00613-f001:**
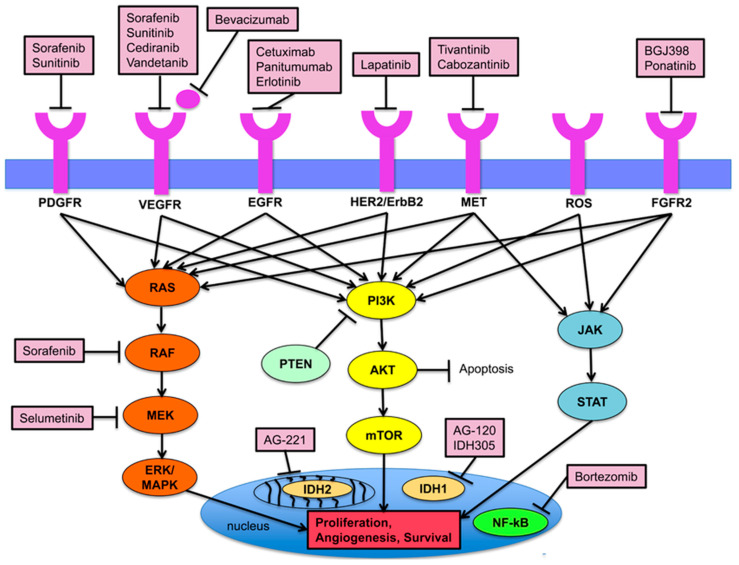
Notable signaling pathways implicated in the progression of cholangiocarcinoma. Used with permission from [5].

## Data Availability

No new data were created or analyzed in this study. Data sharing is not applicable to this article.

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
