# Peer review of "Emerging Therapies in Management of Cholangiocarcinoma"

_cancers, 2024, doi:10.3390/cancers16030613_

Round 1

Reviewer 1 Report

Comments and Suggestions for Authors

The aim of this study was “to comprehensively discuss the latest molecular therapeutics for the treatment of cholangiocarcinoma, including emerging immunotherapies, highlighting promising developments and strategies”.

The review, planned as a demonstration of the latest forms of therapy in cholangiocarcinoma (CCA), essentially fulfills its purpose. It is a pharmacological compendium of the latest drugs approved by the FDA for use or in clinical trials to evaluate their potential use in this cancer.

My comments:

A difficulty for the reader in the text and in Table 1, is the overabundance of proper names of potential therapeutics, components of signaling pathways, or other drugs in combination therapies. This makes the text tiresome, too encyclopedic, and too dictionary-like for a Cancers journal. The One Table has a different way of presenting results over the drug groups described in the text. It is difficult to read. I understand that the criterion of Table 1 phase 1 clinical trials on a given drug, I recommend writing this in the header of the Table. In addition, this Table lacks citations from the literature, even if the data comes from the web. In addition, I recommend supplementing the table with a legend with abbreviations that are missing elsewhere in the text.

There is a lack of summary of knowledge, e.g. which therapies promise the best effects and the least side effects, etc. A summary of the "own research" in this paper should be a figure, e.g., with the signaling pathways that are the most common "targets" for treatment and the molecular therapies used in CCAs, or planned for the future. I believe that a figure should be added to increase the merit of the paper and to provide quicker insight into the authors' acquired knowledge of modern therapies in CCAs.

So paper can be published after the additions that I have indicated.

Author Response

Reviewer 1: 

1. “The One Table has a different way of presenting results over the drug groups described in the text. It is difficult to read. I understand that the criterion of Table 1 phase 1 clinical trials on a given drug, I recommend writing this in the header of the Table.”

Response: We appreciate the reviewer’s excellent comment. The following header has been added to the Table to introduce the contents and to clarify the Table’s organizational structure: “Clinical trials for cholangiocarcinoma (many of which include other biliary tract cancers and solid tumors in their study populations) are listed here, organized by clinical research phase and ClinicalTrials.gov identifier. Advanced disease refers to locally aggressive, metastatic, and/or surgically unresectable tumors.”

2. “In addition, this Table lacks citations from the literature, even if the data comes from the web.”

Response: We thank the reviewer for the comment. Appropriate citations have been added to all clinical trials included in the Table and can be found in the References.

3. “In addition, I recommend supplementing the table with a legend with abbreviations that are missing elsewhere in the text.”

Response: We appreciate the reviewer’s comment. The following legend has been added to supplement the table, clarifying abbreviations that are not found elsewhere in the manuscript: “Legend: ATR kinase, Ataxia telangiectasia and Rad3-related kinase; BTC, biliary tract cancer; CAR, chimeric antigen receptor; HER2, human epidermal growth factor receptor-2; PARP, poly (ADP-ribose) polymerase; SBRT, stereotactic body radiation therapy; TRK, tropomyosin receptor kinase; TRX-MP, troxacitabine monophosphate.”

4. “There is a lack of summary of knowledge, e.g. which therapies promise the best effects and the least side effects, etc. A summary of the "own research" in this paper should be a figure, e.g., with the signaling pathways that are the most common "targets" for treatment and the molecular therapies used in CCAs, or planned for the future. I believe that a figure should be added to increase the merit of the paper and to provide quicker insight into the authors' acquired knowledge of modern therapies in CCAs.”

Response: We appreciate the suggestion to enhance the paper by including a figure summarizing knowledge on common mutations in CCA and corresponding therapies. A figure has been added with permission that highlights different signaling pathways involved in the pathogenesis of CCA and the names of corresponding therapies. This visual representation should provide readers with an overview of modern therapies in CCA.

Reviewer 2 Report

Comments and Suggestions for Authors

I think this is a nice and topic review article regarding latest molecular therapeutics for cholangiocarcinoma. This is well-written and understandable for readers. However, representative pathological/immunohistochemical photo(s) of cholangiocarcinoma or target molecule(s) are needed for differentiation from other adenocarcinomas.

Author Response

  1. “However, representative pathological/immunohistochemical photo(s) of cholangiocarcinoma or target molecule(s) are needed for differentiation from other adenocarcinomas.”

Response: We appreciate the suggestion to incorporate a figure and have addressed this concern by adding a visual representation of signaling pathways involved in the pathogenesis of cholangiocarcinoma. We hope this addition provides a clearer representation of the different molecules involved in the pathogenesis of CCA and how the therapies discussed in this paper relate to these signaling pathways.

Reviewer 3 Report

Comments and Suggestions for Authors

There are many reviews like this article. In addition, there will be only a few agents as alternatives from many ongoing clinical trials. enumeration of clinical trials and descriptions about every agent are not comprehensive.

Author Response

1. “Enumeration of clinical trials and descriptions about every agent are not comprehensive.”

Response: We thank the reviewer for the comment. Appropriate changes have been made to address the reviewer’s comment. Citations have been added to all clinical trials included in the Table and can be found in the References of the manuscript. Additional clarifying information regarding clinical trials has been added to the header and legend of the Table: “Clinical trials for cholangiocarcinoma (many of which include other biliary tract cancers and solid tumors in their study populations) are listed here, organized by clinical research phase and ClinicalTrials.gov identifier. Advanced disease refers to locally aggressive, metastatic, and/or surgically unresectable tumors,” and “Legend: ATR kinase, Ataxia telangiectasia and Rad3-related kinase; BTC, biliary tract cancer; CAR, chimeric antigen receptor; HER2, human epidermal growth factor receptor-2; PARP, poly (ADP-ribose) polymerase; SBRT, stereotactic body radiation therapy; TRK, tropomyosin receptor kinase; TRX-MP, troxacitabine monophosphate.”

2. Concerns with English readability and correctness 

Response: In response to this reviewer’s concerns with English language readability and correctness, we employed grammar editing software to ensure proper English language usage. 

Round 2

Reviewer 3 Report

Comments and Suggestions for Authors

The  authors have revised their manuscript along with reviewers' suugestions.